# Sidescan Only Neural Bathymetry from Large-Scale Survey

**DOI:** 10.3390/s22145092

**Published:** 2022-07-06

**Authors:** Yiping Xie, Nils Bore, John Folkesson

**Affiliations:** Robotics, Perception and Learning Laboratory, Royal Institute of Technology, SE-100 44 Stockholm, Sweden; nbore@kth.se

**Keywords:** bathymetric maps, neural nets, representation learning, sidescan sonars

## Abstract

Sidescan sonar is a small and low-cost sensor that can be mounted on most unmanned underwater vehicles (UUVs) and unmanned surface vehicles (USVs). It has the advantages of high resolution and wide coverage, which could be valuable in providing an efficient and cost-effective solution for obtaining the bathymetry when bathymetric data are unavailable. This work proposes a method of reconstructing bathymetry using only sidescan data from large-scale surveys by formulating the problem as a global optimization, where a Sinusoidal Representation Network (SIREN) is used to represent the bathymetry and the albedo and the beam profile are jointly estimated based on a Lambertian scattering model. The assessment of the proposed method is conducted by comparing the reconstructed bathymetry with the bathymetric data collected with a high-resolution multi-beam echo sounder (MBES). An error of 20 cm on the bathymetry is achieved from a large-scale survey. The proposed method proved to be an effective way to reconstruct bathymetry from sidescan sonar data when high-accuracy positioning is available. This could be of great use for applications such as surface vehicles with Global Navigation Satellite System (GNSS) to obtain high-quality bathymetry in shallow water or small autonomous underwater vehicles (AUVs) if simultaneous localization and mapping (SLAM) can be applied to correct the navigation estimate.

## 1. Introduction

High-resolution bathymetry is valuable for various underwater applications. Traditionally multi-beam echo sounder (MBES) is used to collect bathymetric data due to its ability to return three-dimensional measurements thanks to its two perpendicular arrays. Sidescan sonar (SSS) with its single linear transducer array, on the other hand, can not be directly used to reconstruct the seabed’s geometry. However, its returns can be well approximated by a Lambertian model which makes it possible to use shape-from-shading (SFS) techniques [1] to reconstruct bathymetry from sidescan images. Reconstructing bathymetry from a single SSS line is a well-known ill-posed problem but combining multiple SSS lines and leveraging recent advances in Deep Learning make the problem possible under certain assumptions. However, one challenge is that the returned intensity is not only a function of the surface normal but also affected by the seabed reflectivity. One approach is to classify the seabed sediments from the collected sidescan data (via objective or subjective analysis [2]), while another is to estimate the seabed reflectivity directly without determining the sediment types explicitly [3,4].

Although the new technological advances on phase differencing sonars, also known as interferometric sonars (e.g., Edgetech 6205 from EdgeTech, United States and Klein HydroChart 3500 from KLEIN, MIND Technology, United States), could acquire bathymetric measurements, which could be of great help to many applications [5], they also have some restrictions. For example, the Edgetech 6205 is mainly suitable for shallow water, less deep than 35 m [6]. Another restriction is that the measurements of interferometric SSS are noisier containing many outliers that must be filtered out [7]. However, in this work we focus on using non-interferometric SSS for the following reasons: they are ubiquitous in the AUV community and provide a more affordable solution requiring less power.

Many previous works have proposed different methods to reconstruct bathymetry from sidescan [3,4,8,9,10,11,12,13,14,15,16,17]. They can be categorized into different groups according to different criteria, see Table 1. In terms of whether the external bathymetric data are required, refs. [8,9,10,11,12,13,14] need external bathymetric data, either from sparse direct bathymetric measurements [8,10,11,12,13,14] or coarse multi-beam data [9]. On the other hand, refs. [3,4,15,16,17] do not require external bathymetric data where [15] assume the altitude of the towfish is known and reconstruct the relative shapes of the seafloor with a linear method using simulated sidescan. Refs. [3,4] use a flat seafloor assumption as the initialization and [16,17] use the first bottom return in the sidescan waterfall images to obtain the altitude and by combining data from the pressure sensor to get sparse bathymetric information. Note that both [15,17] use actual seabed data and simulated sidescan from the actual seabed to assess their proposed methods. In terms of how to model the scattering process, most of the previous works use the Lambertian model [8,9,10,12,14,15,16,17] while [11,13] use data-driven approaches, i.e., deep neural networks for the scattering modeling.

This work also uses the Lambertian model for the scattering modeling, whose advantage over using a data-driven model is that it does not require the “ground truth” bathymetry to create the training data as was needed in [11,13]. In addition, the accurate registration between the ground truth bathymetry formed from MBES data and sidescan is crucial for a data-driven model, where less accurate registration would result in less accurate estimates [11,13]. That registration can never be perfect in practice as it is impossible to remove small timing issues or to know the true sound velocity profile experienced by each sonar beam.This work is an extension of our previous work [12]. The extension is that we remove the requirement of external bathymetric data by modelling the vertical sidescan sonar beam pattern near the nadir. Naively, the idea is that knowing the beam pattern and seeing the range of the first return closest to the nadir allows a bathymetric estimate that can replace the altimeter values used in our previous work. Therefore, our proposed method can be used in the situation where only sidescan data are available. The altitude of the sonar is implicitly modeled from the nadir of the waterfall images and similar to [17] that will be optimized directly by minimizing the loss on sidescan intensities. The mathematical model in this work is similar to [3,4] but we use a neural network as the bathymetric representation instead of a grid as most of the previous work or triangle meshes [17]. Unlike [15,17], the method proposed in this work was evaluated on real sidescan data instead of simulated data. In addition, unlike [16] where the reconstructed bathymetry is assessed by comparing to linear interpolation of single-beam echo sounder (SBES) data, we in this work evaluate our methods by comparing the reconstructed map to high-precision MBES bathymetry.

One of the benefits of using implicit neural representations (INR) is that the memory requirement is independent of the spatial resolution due to the neural network representation being continuous. The past few years have seen a rising interest in implicit neural representations or coordinate-based representations to represent a scene rather than explicit representations such as voxels; point clouds and triangle meshes. Different implicit representations have been proposed, such as Occupancy Networks [18], Deep Signed Distance Function (DeepSDF) [19], Scene Representation Networks (SRNs) [20], Neural Radiance Fields (NeRF) [21], and Sinusoidal Representation Networks (SIRENs) [22]. Among these, Occupancy Networks learn a binary classifier and use its decision boundary to represent the scene. NeRF is more suitable for applications such as novel views synthesizing. SIRENs outperform ReLU-based Multi-layer Perceptrons (MLPs) such as DeepSDF and SRNs in terms of capturing details when modeling large scenes [22], since the sine activation functions make SIRENs be able to create high quality gradients of the represented signals.

The overview of the proposed method (we plan to release the code in the future) is shown in Figure 1. The estimated bathymetry is parameterized by a neural network that takes the Euclidean easting and northing coordinates as input and predicts the corresponding seafloor height estimates. Given a sidescan ping, the sonar’s position and orientation are used to calculate the position of echoes on the seafloor. The surface normal at that point can be calculated using the gradient of the seafloor, which then can be used in a Lambertian scattering model to approximate the returned intensity. By comparing the measured intensity and the approximated intensity, the gradient of the intensity loss can be backpropagated to update the neural bathymetry. Note that the loss is calculated on all the intensities, including the nadir area, which indicates the altitude of the sonar, allowing the constraints from the sidescan itself to estimate the relative seafloor height. Combing the readings from the pressure sensor, we can estimate the absolute seafloor height from sidescan without external bathymetric data.

The main contribution here is that the proposed method could produce high-quality bathymetry with sidescan data only, thus having the potential of using smaller AUV platforms with relatively simple equipment to carry out high-quality bathymetric surveys. We also validate that our neural rendering method is capable of generating these maps given good navigation and a sidescan sonar alone.

## 2. Methods

### 2.1. Sidescan Sonar Overview

Figure 2 illustrates the rear view and the top view of a sidescan at $O(O_{x},O_{y},O_{z})$ operating at altitude *h*. Sidescan emits a sound pulse to ensonify a swath of the seafloor and collect the reflected returns and their travel times. The sonar will not get echo returns from the seafloor until the sound passes through all the water columns, leaving a distinct separation in the sidescan waterfall image (Figure 3). The first return from the seabed $p_{fbr}$ is call the first bottom return, which is very useful to determine the altitude of the sonar. Typically the sonar’s vertical beam width $\alpha_{v}$ is about 40–$60^{\circ }$ and the horizontal beam width $\alpha_{h}$ is about $1^{\circ }$. For a point $p(P_{x},P_{y},P_{z})$ on the seafloor within the ensonified swath, it can be expressed by its polar coordinates $\left(d,\phi_{g}\right)$, where *d* is the distance between point *p* and the sonar, often referred to as the *slant range* and $\phi_{g}$ often referred to as the *grazing angle*. For a sidescan, the slant range *d* is known but not the grazing angle $\phi_{g}$ due to the unknown seafloor underneath. That leaves an ambiguity in the angles in the yz-plane, making point *p* can be at any grazing angle $\phi_{g}$ within $\alpha_{v}$ along the arc at distance *d* from the sonar assuming isovelocity sound velocity profile (SVP). Another ambiguity for sidescan is in the xy-plane, where the ensonified point *p* is ambiguous over the arc *q* due to the sensor opening in the xy-plane $\alpha_{h}$. However, most of the time this can be ignored due to the fact that $\alpha_{h}$ is usually very small.

### 2.2. Neural SFS

In this section, we will define our bathymetric representation and how to simulate sidescan intensities using such representation in a differentiable manner (see Table 2 for the inputs and outputs of the proposed method). At last, we will describe how to optimize the system to obtain the estimated bathymetry.

#### 2.2.1. Implicit Neural Representation

We use an implicit neural representation (INR) for the bathymetry, i.e., a function $\theta :R^{2}\to R$ that maps Euclidean easting and northing x,y to its corresponding seafloor height $\tilde{h}$. The implicit neural representation θ is parameterized by a variant of a MLP, known as the SIREN [22], the same as in our previous work [12,13]. Although θ can, in principle, be any differentiable neural network, SIREN has the advantage of producing high-quality first-order derivatives [22], allowing to reconstruct small features of the seafloor. Note that θ is differentiable, which is crucial to optimize the bathymetry given losses on sidescan intensities. In addition, differentiability implies continuity, meaning that the size of the neural network model θ is independent of the spatial resolution of the bathymetry, which is one of the advantages of INR over explicit methods such as triangle meshes.

#### 2.2.2. Sidescan Scattering Model

Similar to our previous work [12], the proposed method in this work requires precise positioning, which could be available from a surface vessel. In addition, we assume the horizontal beam width $\alpha_{h}$ (see Figure 2) is narrow enough to be neglected, thus leaving the ensonified volume for every ping *i* lying in one plane. For every bin *n* in ping *i*, assuming isovelocity SVP, the ensonified volume $p_{i,n}\left(\phi \right)$, is at a fixed distance $d_{n}$ away from the sonar; parameterized by angle ϕ, where $\phi =90^{\circ }-\phi_{g}$ and $\phi_{g}$ is the grazing angle in Figure 2. As aforementioned, ϕ is unknown without knowing the bathymetry. However, given the estimated bathymetry θ, we can use gradient descent to determine $\phi_{i,n}$, as shown in Figure 4 (assuming facing starboard). Once the seafloor intersection is found, we can define the surface normal at point *p* with respect to θ, given the gradient components $\nabla_{x}$,$\nabla_{y}$: (1)$$N^{\theta }\left(p\right)={\left[-\nabla_{x}\theta \left(p_{x};p_{y}\right),-\nabla_{y}\theta \left(p_{x};p_{y}\right),1\right]}^{T}.$$

Using the FLD sensor frame (see Figure 4), assuming the sonar facing the starboard side, the rotation from the tangent to the surface normal is: (2)$$r_{i}\left(\phi_{i,n}\right)=d_{n}R_{i}{\left[0,\sin\left(\phi_{i,n}\right),cos\left(\phi_{i,n}\right)\right]}^{T}.$$

Given $N^{\theta }\left(p\right)$ and $r_{i}\left(\phi_{i,n}\right)$, we can use the Lambertian scattering model to compute the cosine of the incidence angle. In particular, we use the $cos^{2}$ approximation [15] same as in our previous work [12] due to giving better approximations at low grazing angles: (3)$$M_{i,n}^{\theta }\left(\phi_{i,n}\right)={\left(r_{i}{\left(\phi_{i,n}\right)}^{T}N^{\theta }\left(p\right)\left(p_{i,n}\left(\phi_{i,n}\right)\right)\right)}^{2}.$$

In addition, we also estimate the beam profile, gain and the albedo jointly. Similar to [12], we estimate the gain $A_{i}$ for each sidescan line. The albedo of the seafloor for the whole surveyed area R(p) is a radial basis function parameterized by kernel weights $R_{K}$, which are learned during the optimization. The beam profile Φ(ϕ) is also defined as a radial basis function with fixed-position kernels evenly spread across ϕ.

#### 2.2.3. Nadir Model

Our gradient descent procedure produces for each sidescan intensity the point $p_{i,n}\left(\phi \right)=\left(p_{x};p_{y};p_{z}\right)$ that is closest to the seafloor on the corresponding arc. If an arc covers a volume that is completely in the water column above the seafloor, our gradient descent procedure will not find an intersection with the seafloor, instead producing the closest value above it, $p_{z}>\theta \left(p_{x};p_{y}\right)$. In our sonar model, such bins correspond to low-value intensities within the nadir area. To reflect that within our model, we employ a smooth indicator function that ensures that the intensity has no value if the closest value is far from the seafloor, and has a value close to one if intersecting the seafloor. If we denote the signed offset
(4)$$\Delta^{\theta }\left(p\right)=\theta \left(p_{x};p_{y}\right)-p_{z},$$
the indicator is given by the kernel: (5)$$\delta_{i,n}^{\theta }\left(\phi \right)=\exp\left(-\frac{\Delta^{\theta }{\left(p_{i,n}\left(\phi \right)\right)}^{2}}{\sigma_{N}^{2}}\right).$$

The spread parameter $\sigma_{N}$ can be manually tuned to fit the smoothness of the nadir line in the measured data.

Putting everything together, our full sonar model is given by the product of the scattering model from [12] with our novel nadir model,
(6)$${\tilde{I}}_{i,n}=KA_{i}\delta_{i,n}^{\theta }\left(\phi_{i,n}\right)M_{i,n}^{\theta }\left(\phi_{i,n}\right)\Phi \left(\phi_{i,n}\right)R\left(p_{i,n}\left(\phi_{i,n}\right)\right).$$
where *K* is the normalization constant. Notably, modeling the nadir transition enables us to model the full sidescan range, as opposed to just the outer region past the nadir. In addition, the information gained from the nadir should enable the network to implicitly reason about the seafloor depth.

#### 2.2.4. Optimization

From our previous work (Section III-B) [12], having one normalization constant *K* estimated for the whole dataset prior to training can improve convergence. The loss function is merely minimizing the measured sidescan intensities and the simulated intensities: (7)$$L=\frac{1}{\left|\left\{I_{i,n}\right\}\right|}\sum_{I_{i,n}}∥\;{\tilde{I}}_{i,n}-I_{i,n}\;∥$$
where $\left|\cdot \right|$ denotes the size of the set $\left\{I_{i,n}\right\}$ and $∥\cdot ∥$ denotes the $L_{2}$ norm. It is worth mentioning that the gradient descent procedure to find seafloor intersections is not differentiable; therefore, we compute the intensity in Equation (Equation 6) without back-propagating through the gradient descent procedure, similar to our previous work (Section II-E) [12]. Our previous work showed that its impact could be neglected once θ has converged to the range of the true height.

#### 2.2.5. Assessment

To assess the proposed method, we evaluate the reconstructed bathymetry using mean absolute error (MAE), maximum error, minimum error and standard deviation of the errors compared to the high-resolution MBES bathymetry. The standard deviation is calculated as: (8)$$STD=\sqrt{\frac{\sum_{i=1}^{N}{\Delta Z}^{2}}{N}},\Delta Z=\theta \left(p_{x},p_{y}\right)-p_{z,gt},$$
where $\theta (p_{x},p_{y})$ is the estimated depth point and $p_{z,gt}$ is the ground truth value obtained from MBES, and *N* is the number of points in the height map.

## 3. Experiments

### 3.1. Data and Surveyed Area

The data to assess the proposed method are collected with a nearshore surface vessel called *Ping* (see Figure 5). *Ping* is equipped with a Real-Time Kinematic (RTK) GPS, ensuring the high precision of positioning. It also has a dual-head Reson 7125 MBES which collects the bathymetric data that are used only for evaluation. The Edgetech 4200 MP sidescan sonar is hull-mounted, ensuring sonar’s attitudes are accurate. The surveyed area contains large features such as a hill, a ridge and some rocks. The area is at about 9–25 m depth. Further details on the sensor and survey area are presented in Table 3.

The whole dataset contains 52 surveying lines of sidescan and the sidescan sonar Edgetech 4200 MP has a 900 kHz sampling frequency with 1 cm across-track resolution and 0.18 m along-track resolution. The raw XTF files are parsed (see Figure 3) using an open-source software, auvlib [24], developed by ourselves and the waterfall images are downsampled to have 64 bins per side for training, which were found empirically to have a good trade-off between being able to reconstruct some of the rocks in the final bathymetric map and the training time (11–12 h). Three examples of the downsampled waterfall images can be seen from Figure 6, where each waterfall image contains 500 pings and each ping has 128 bins for both port and starboard side.

### 3.2. Training Details

Similar to our previous work [12], we use a 5-layer MLP with the hidden layer size of 128, where the activation function is the sine function. All positional data are normalized to the range [−1,1]. We train the MLP for 800 epochs using an Adam optimizer with a learning rate of $2\times 10^{-4}$ decayed by the factor 0.995 after every epoch. For each batch we random sample 100 sidescan pings, each of which has 128 bins, 64 per side. We iterate through all the sidescan data (roughly 93,000 pings) for one epoch. The whole training takes approximately 12 h on a single Nvidia GEFORCE RTX 2080 Ti GPU. Note that the estimated bathymetric map converges pretty well in a large scale at 400 epochs, though more epochs with a smaller learning rate would help the reconstruction of smaller details such as rocks. The beamform Φ(ϕ) is only estimated for angles ϕ within $80^{\circ }$ for either side, based on our knowledge of the tilt angle $\theta_{t}$. In particular, we use 20 fixed-position kernels across $\left[0^{\circ },80^{\circ }\right]$ but set the kernel weights corresponding to angles within $\phi =10^{\circ }$ to be a constant value *C*. This is due to the fact that γ in Figure 2 can not be ignored and given our knowledge it should be around $20^{\circ }$.

## 4. Results

We assess our proposed method by comparing the estimated bathymetry to the high-precision bathymetry from MBES with a resolution of 0.5 m. We use several metrics to evaluate the quality of reconstructed bathymetry, mean absolute error (MAE), maximum, minimum and standard deviation of the errors of the bathymetric map and the cosine similarity [25] (bounded in [0,1], 1 being identical) of the gradients of the bathymetric map, similar to our previous work [12].

We begin with the qualitative results of the estimated bathymetry. Figure 7 shows the comparison between the ground truth bathymetry and the one estimated from sidescan. In addition, the gradients of the bathymetric maps are displayed in Figure 8, which shows that the proposed method manages to reproduce not only the large-scale features such as the hill and ridges but also some of the smaller features such as rocks. However, there are many small rocks that are not reconstructed.

To assess the estimated bathymetry further we calculate the errors on the bathymetry for the proposed method and compare them with previous works [12,16], see Table 4. The estimated bathymetry of the proposed method has a bias of around 10 cm and an average error of 20 cm with standard deviation of 23 cm. However, the maximum and minimum errors are a lot larger than the mean absolute error of 20 cm. The probability distribution function (PDF) curves of the error is shown in Figure 9. We also compare the performance using our previous method [12] and method used in [16] in Table 4. Note that our previous method [12] used the MBES to simulate the altimeter to provide external bathymetric measurement so that the mean absolute error and standard deviation are at centimeter level. In addition, the results from [16] are compared with the Digital Elevation Model (DEM) constructed from SBES and the surveyed area has less variation on the terrain (from 10–13 m depth), while [12] and this work are assessing the proposed methods with data collected from high-resolution MBES from an area at 9–25 m depth.

The albedo and beamform are also estimated with the bathymetric map at the same time. The estimated albedo (see Figure 10) seems to have higher coefficients in the hill and rocky areas, which is expected. However, the upper left corner area also has higher albedo coefficients which should be related to the artifacts in the same area in Figure 8.

The estimated beamform is displayed in Figure 11 and Figure 12, where we can see the estimated beam profile matches the analytical model [26] using a linear-phased array with a tilt angle $\theta_{t}=40^{\circ }$. The theoretical beam pattern is calculated as: (9)$$\Phi_{l}\left(\phi \right)={\left(\frac{sin(ksin\left(\phi -\phi_{0}\right))}{ksin(\phi -\phi_{0})}\right)}^{4},$$
where k=3.29 for a (one-way) 3 dB beam width of $50^{\circ }$ and $\phi_{0}$ is half of the beam width.

When estimating the beamform, we set beamform corresponding to ϕ under $10^{\circ }$ to a constant value. Doing such considerably helps the convergence of the optimization. We found out that using different values for the constant *C* did not affect the quality of the gradient maps, as shown in Figure 13a. The qualitative results for different *C* are displayed in Appendix A, from which one could barely tell the difference from each other. However, as one can see from Table 5, the absolute height map could deviate from the actual bathymetry, also shown in Figure 13b. This shows the proposed method could estimate the relative height pretty well from sidescan data itself, but one might argue that giving only a few points of absolute height of the seafloor (or even one) could “pin down” the seafloor and reduce the absolute error.

Given estimated bathymetry θ, estimated albedo *R* and estimated beam profile Φ, we can calculate the simulated sidescan intensities given sonar’s attitude. An example (more in Appendix B) is displayed in Figure 14 where on the left is the measured waterfall image downsampled to 64 bins per side and on the right is the simulated SSS. The simulated SSS can be georeferenced using the estimated bathymetry, as shown in Figure 15. We can also find that using a Lambertian model gives an image with much less noise. This suggests that some speckle noise filtering on the measured SSS with a higher resolution (or even without downsampling) might further improve the estimated bathymetry, especially for the small features such as rocks.

## 5. Discussion

### 5.1. External Bathymetric Measurement

In the proposed method, the absolute depth of the seafloor is obtained by combining the depth of the sonar and the altitude implicitly learned from the nadir model (Equation (Equation 5)). With the novel nadir model and modelling the full beam pattern, we could remove the requirement of external bathymetric measurement in our previous work [12]. In particular, our previous work [12] used the MBES measurement to simulate the altimeter readings due to lacking of an extra altimeter sensor on the surveying vehicle. However, with the new nadir model, we could in theory use the same vehicle to survey a new area of interest with no prior map and use sidescan data to create a bathymetric map using the proposed method.

Since we do not have a separate altimeter sensor reading in this particular dataset, we can only compare the performance between using a “perfect” altimeter [12] and using first bottom return to extract altitude in Table 4. Grantedly, using the MBES to simulate the altimeter as external bathymetric measurement would give a better performance, namely lower average error and standard deviation.

### 5.2. Implicit Neural Representations: Sirens

The proposed method uses Implicit Neural Representations, in particular a SIREN to represent the bathymetry. Implicit Neural Representations could potentially be more memory efficient than explicit representations (e.g., triangle meshes). In addition, compared to implicit surface representations (e.g., radial basis functions), SIRENs make fewer assumptions by leveraging learning from data and can produce higher quality gradients [22] because of the sine activation functions. In particular, SIRENs are more suitable to the proposed method since we need access to the gradients of the seafloor for the Lambertian model, compared to the MLPs with ReLU activation functions with sinusoidal positional encoding [21].

### 5.3. Shadows

We assume the shadows can be neglected due to there being few shadows in our dataset specifically. Additionally, because of the downsampling procedure, very few pixels in the waterfall images correspond to shadows and we empirically found out that simply ignoring the shadows did not affect the convergence of the model. They are possibly ignored during the optimization as if they are outliers. Another way of dealing with shadows is to empirically set a threshold to remove all the points corresponding to them, as we did in our previous work [12]. However, shadows contain some information about the seafloor features and one way to model the shadows is through differentiable rendering [17], where the height of the protrusions causing the shadows can be inferred.

### 5.4. Noise in SSS Data

We, in this work, use downsampled SSS to reconstruct the bathymetry, which is conducted by averaging the SSS intensities within larger intervals. Such a procedure has weakened the effect of noise in the SSS images. However, if SSS data contain echoes from objects, fish or abnormal noise in the water column (see Figure 6c), the estimated bathymetry will still be affected. However, one of the disadvantages of downsampling is losing some useful information. The alternative of using downsampling to reduce the assumed white noise in the SSS waterfall images is to properly model the noise for underwater acoustic signals [27,28] and transform the noise distribution from acoustic signal processors to the SSS images.

### 5.5. SSS Resolution

In theory, the proposed method can be used on the raw sidescan images since the method does not rely on downsampling. Downsampling SSS reduces some of the noise but also reduces the across-track range resolution and loses some information. The across-track range resolution, in particular, would limit how accurate the altitude inferred from the first bottom return of SSS can be. Increasing the SSS resolution might be able to reproduce very small features of the seafloor, e.g., small rocks. However, using raw sidescan pings to train might run into some practical issues, such as taking too much memory so that each batch can only contain very few pings. In addition, the positioning error and orientation error will most likely be a problem at this level of resolution, especially at the outer part of the sidescan pings.

### 5.6. Refraction

In this work, we ignore the refraction of sound waves treating the sound ray as a straight line and we also assume isovelocity, which would result in an inaccurate calculation of the return echo’s position. This effect would be more severe using higher SSS resolution or surveying a deeper area.

### 5.7. Potential Applications: Small AUV Bathymetry

Bathymetry surveys have gone from lead and line to MBES. These MBES originally were attached to ships, but now are beginning to be used from AUVs. Commercial surveys use very large and expensive AUVs such as the Kongsberg Hugin. These AUVs are equipped with excellent MBES, inertial navigation, and Doppler Velocity Log. Generally a large size is needed to mount a high quality MBES and to have enough thrust to overcome any ocean currents with minimal navigation error. They can travel long missions close to the seafloor with relatively low drift in dead-reckoning estimates. By being closer to the seafloor the horizontal resolution is improved while the swath width is decreased. The survey can be programmed and after the AUV is launched the ship need only be available for emergency ascents if they occur and for the recovery at the end of the mission. It is the high resolution and autonomy from the ship and crew that is attractive.

However there are drawbacks. These AUVs are difficult to launch and recover, requiring special rigging and a large ship. The decreased swath increases the time for the same coverage. There also is the problem of geo-referencing the data without any GPS. The AUVs themselves are very costly and could be lost.

If one could extract the same or better bathymetry and navigation from sidescan sonar collected from a small AUV a far larger swath could be covered with far lower requirements on the size of the ship. If enough features are present on the sea floor simultaneous localization and mapping, SLAM, techniques could be applied to the sidescan images to post-process the dead-reckoning and sonar to give a corrected navigation estimate, Figure 16. The neural rendering based methods we are working on and presented here have the promise to provide a bathymetric estimate at the cross-track resolution of the sidescan i.e., a few cm. The longitudinal resolution would depend on speed and ping rate (related to slant range). So far we have looked at how the method can indeed provide good bathymetry. As the models are fully differentiable w.r.t all parameters including the AUV positions and orientations as well as the bathymetry, we could in principle use the same framework to refine the navigation estimate if started from a sufficiently good SLAM approximation. We might iterative correct the bathymetry then the navigation only to refine the bathymetry again.

A drawback to using smaller AUVs is that they would be slow. In addition, even if the individual swath of the sidescan is very large there needs to be many overlapping swaths from a few different AUV headings for the estimates to correctly converge. On the other hand while handling multiple Hugin AUVs is extremely difficult, launching many hand held AUVs is not. We therefore propose a team of AUVs launched together to carry out a survey in unison. Thus we can collect multiple overlapping swaths in one pass, then repeat from a second or third heading. Equipping the AUV with acoustic modems and time synchronizing them would allow one AUV to ping the others, which when comparing the send and receive times post mission would allow the relative ranges between the pinging AUV and the others to be determined. This would then be an additional constraint that could be added to the SLAM optimization framework. This is likely needed to replace the dead-reckoning between lines one would have when surveying with on AUV over a longer time. By alternating the pinging AUV a web of constraints could be formed between the AUV trajectories. This vision is illustrate in Figure 17.

## 6. Conclusions

We present a neural shape-from-shadow method that can be used to reconstruct bathymetry from sidescan sonar data alone without external bathymetric data. We demonstrate in our experiments that the proposed method can fuse multiple lines of sidescan data from a large-scale survey into a self-consistent bathymetric map through global optimization. The assessment of comparing with the bathymetric map constructed from a MBES shows quantitative errors but gives a qualitatively good estimate of the relative heights of the seafloor, clearly indicating the large features such as hills, ridges and large rocks. The proposed method has wide applications for small surface vehicles equipped with GNSS or small underwater vehicles if SLAM can be used to obtain accurate positioning, which usually lack large MBES sensors.

Future work should focus on using a higher resolution of SSS while reducing the noise so that smaller features of the seafloor can be reconstructed. In addition, one might want to take the sonar’s attitudes as parameters into optimization jointly with the bathymetry, instead of treating them as a given. Another possible work is to combine raw SSS and MBES data to produce a super-resolution bathymetry to reconstruct micro-features that are only visible from SSS images.

## Figures and Tables

**Figure 1 sensors-22-05092-f001:**
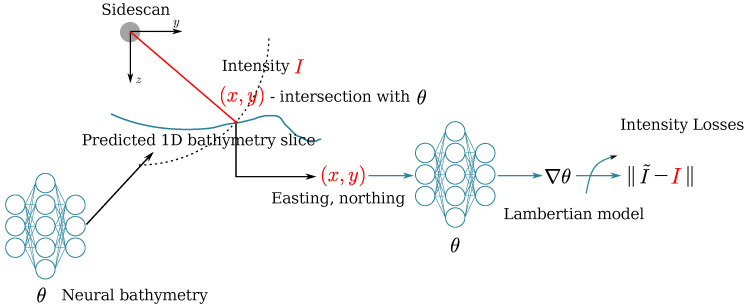
Overview pipeline of the proposed method. The bathymetry is represented by a neural network θ which takes the easting and northing (x,y) as the input and output the corresponding height of the seafloor. Given θ and the sonar’s current attitude we can find the positions of the sonar echoes and the gradients of the bathymetry ∇θ are computed (θ is differentiable) so that we can use a Lambertian model to estimate the corresponding intensities $\tilde{I}$. The measured SSS intensities *I* are used as a reference to compute the loss and we use a gradient-based optimizer to optimize the bathymetry θ so that the simulated intensities match the measured ones.

**Figure 2 sensors-22-05092-f002:**
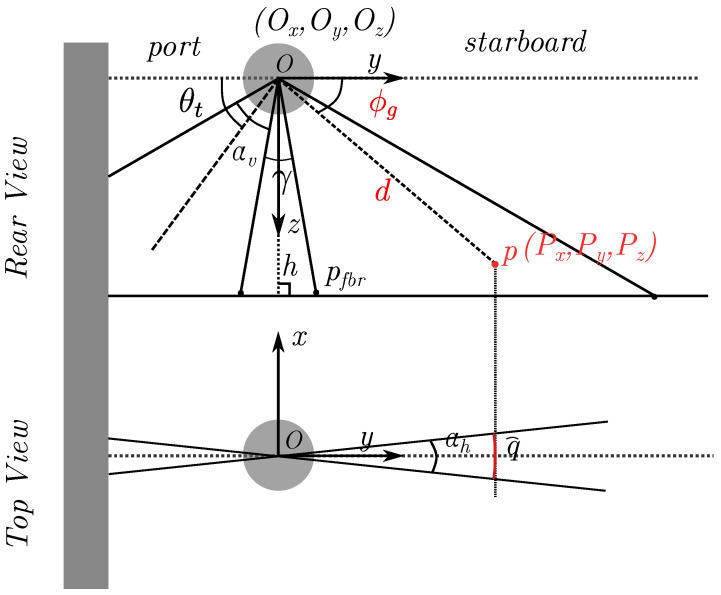
Sidescan sonar illustration in Forward Lateral Down (FLD) frame (modified from [17]). The sidescan sonar usually has two heads symmetrically mounted on the port and starboard side of the AUV with a fixed angle $\theta_{t}$, often referred as to *tilt angle*, which is the angle between the horizontal axis and the center of the vertical beam $\alpha_{v}$. There is usually a gap γ between the two heads leaving a nadir gap where sidescan does not have coverage. Sidescan usually has a wide vertical beam $\alpha_{v}$ (often referred as to the sensor opening in the yz-plane) and very narrow horizontal beam $\alpha_{h}$ (often referred as to the sensor opening in the xy-plane).

**Figure 3 sensors-22-05092-f003:**
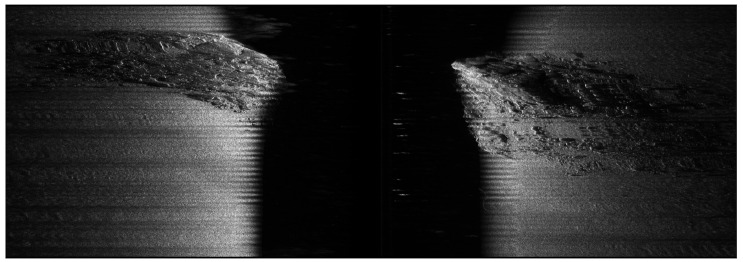
An example of a raw sidescan sonar waterfall image containing 1500 pings with 1 cm across-track resolution and 0.18 m along-track resolution, where the sonar is across a hill with many rocks. The intensity is normalized to the range [0,1] and 1 being the highest echo strength.

**Figure 4 sensors-22-05092-f004:**
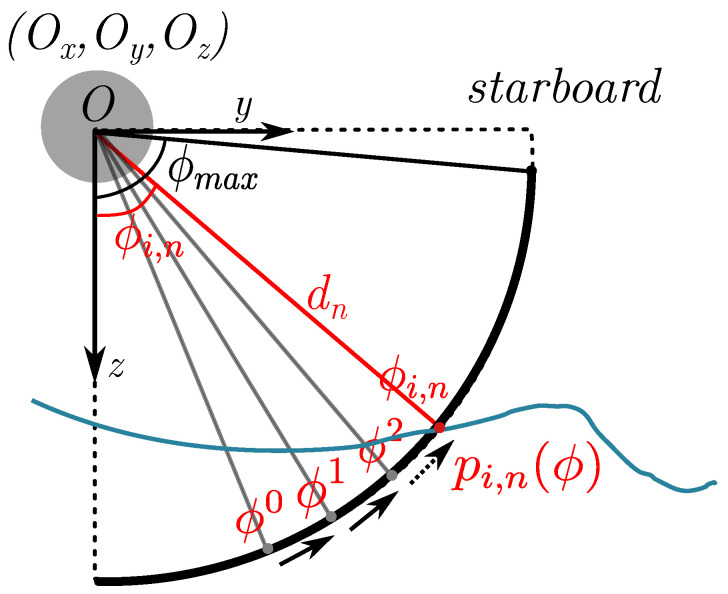
Sidescan sonar frame for the starboard side. The bathymetry θ is illustrated as a 1D slice assuming lying within the sidescan plane. To find the intersection with the echo $p_{i,n}\left(\phi \right)$ parameterized by the angle ϕ, we use a gradient descent algorithm to find $\phi_{i,n}$ starting from $\phi_{0}$.

**Figure 5 sensors-22-05092-f005:**
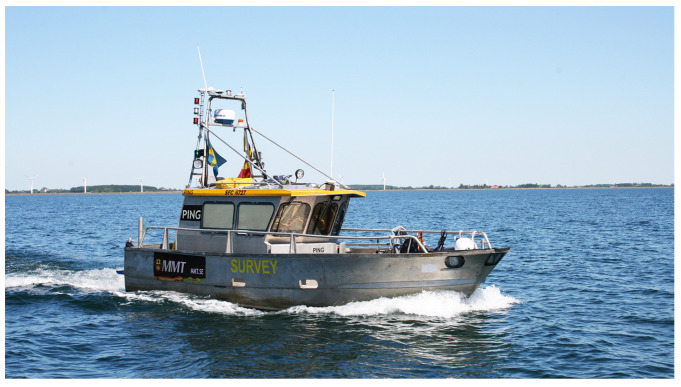
MMT’s nearshore survey vessel Ping [23].

**Figure 6 sensors-22-05092-f006:**
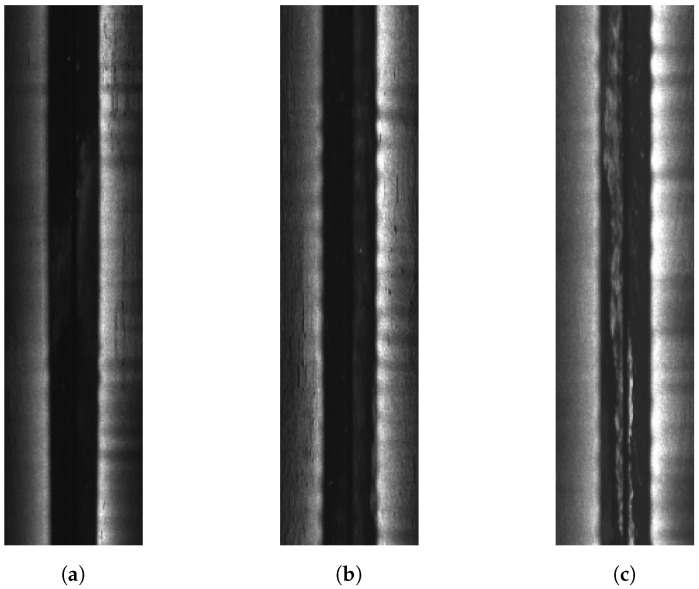
Examples of downsampled sidescan sonar waterfall images, where each waterfall image contains 500 pings and 128 bins (port and starboard). (**a**) Near the rocky area from southwest to northeast. (**b**) Between the hill and the ridge from west to east. (**c**) Flat area from east to west, where there is abnormal noise in the water column. Note that the port side in general appears to have higher intensities than the starboard side.

**Figure 7 sensors-22-05092-f007:**
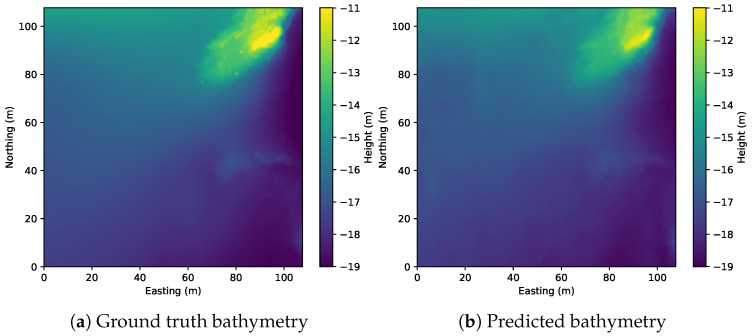
Bathymetric maps for the surveyed area, where (**a**) is constructed from MBES bathymetric data and (**b**) is estimated from sidescan data with a global optimization.

**Figure 8 sensors-22-05092-f008:**
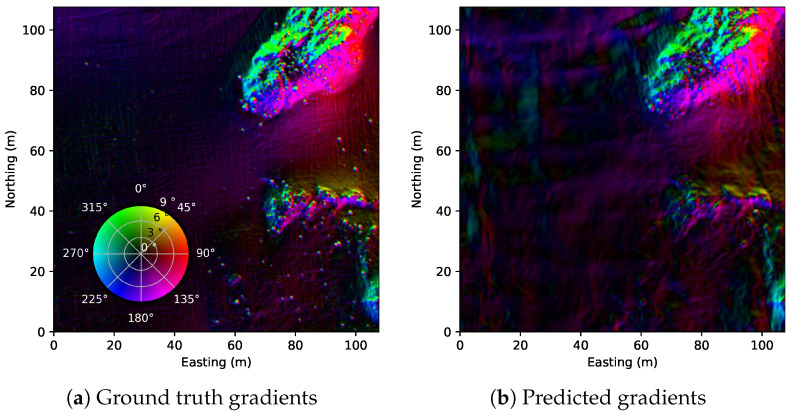
Gradients maps for the surveyed area, where (**a**) is calculated with finite differences and (**b**) is directly calculated from estimated bathymetry since θ is differentiable.

**Figure 9 sensors-22-05092-f009:**
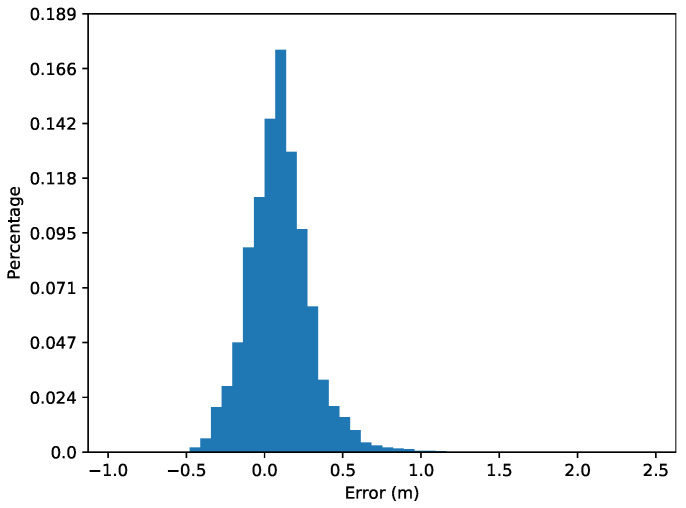
PDF curves of errors of the surveyed area.

**Figure 10 sensors-22-05092-f010:**
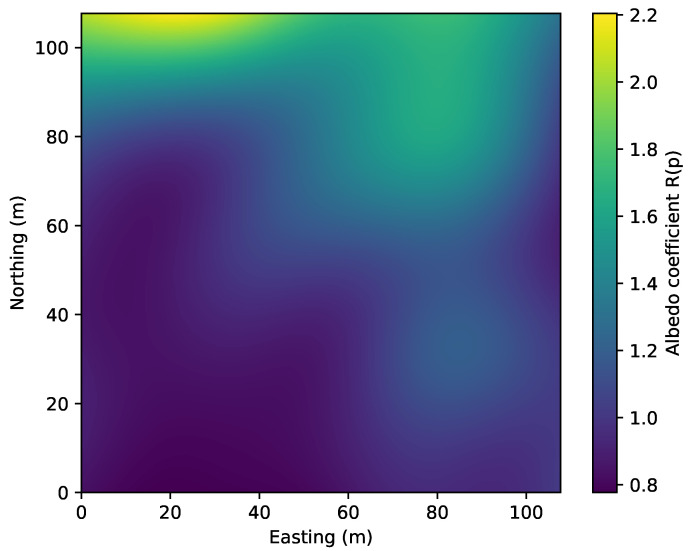
Normalized estimated albedo coefficient across the surveyed area.

**Figure 11 sensors-22-05092-f011:**
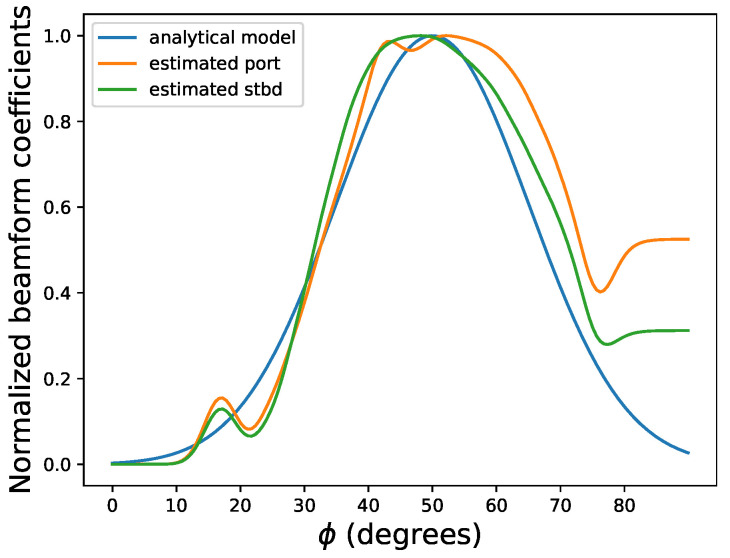
Normalized beamform, where the blue line is using the linear-phased array model to calculate the beamform (at title angle $40^{\circ }$), the orange line and green line are estimated beamform from the optimization for port and starboard, respectively.

**Figure 12 sensors-22-05092-f012:**
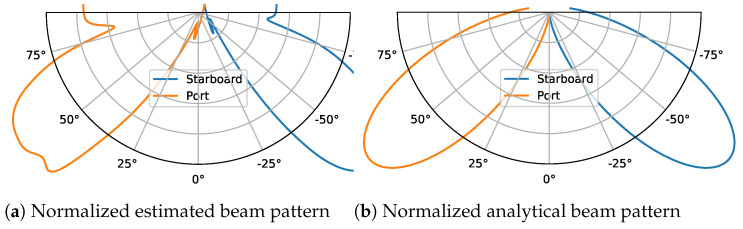
Normalized beam pattern for port and starboard side, (**a**) estimated beam pattern and (**b**) analytical model using a linear-phased array with the center of beam at $\phi =50^{\circ }$. We can notice the effect of setting angles within γ to a constant value in (**a**).

**Figure 13 sensors-22-05092-f013:**
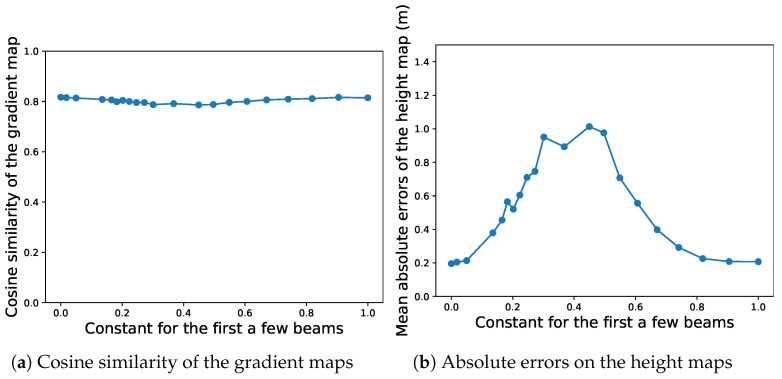
Setting the beamform of the angles within γ to an arbitrary constant value does not affect the convergence of the optimization or the quality of the gradient maps. (**a**) shows that the cosine similarity of the estimated gradients against the ground truth stays steadily around 0.8 as the constant *C* changes from 0 to 1. However, the absolute errors on the bathymetry could deviate from the ground truth from 20 cm to more than 1 m without external bathymetric data as constraints.

**Figure 14 sensors-22-05092-f014:**
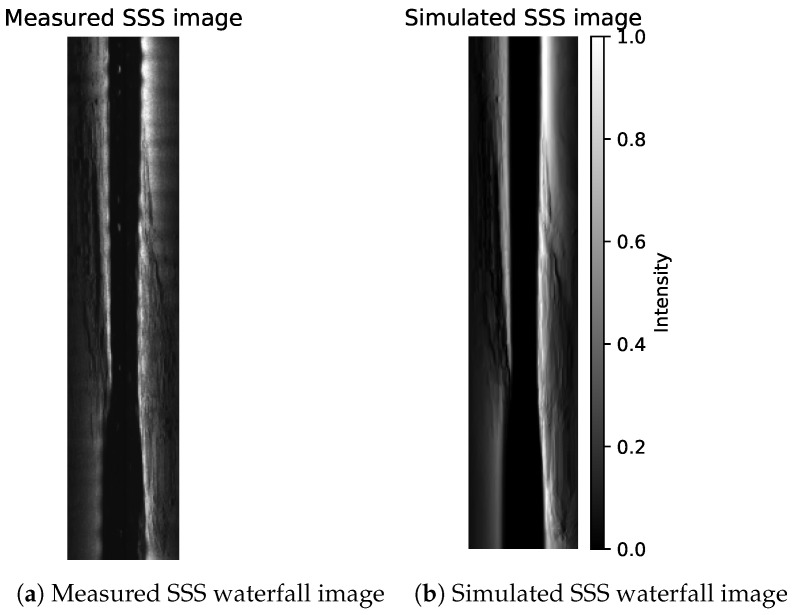
An example of SSS waterfall image containing 600 pings across the hill from east to west. (**a**) Measured SSS downsampled to 64 bins per side for training and (**b**) SSS using the Lambertian model calculated from estimated bathymetry, albedo and beam pattern.

**Figure 15 sensors-22-05092-f015:**
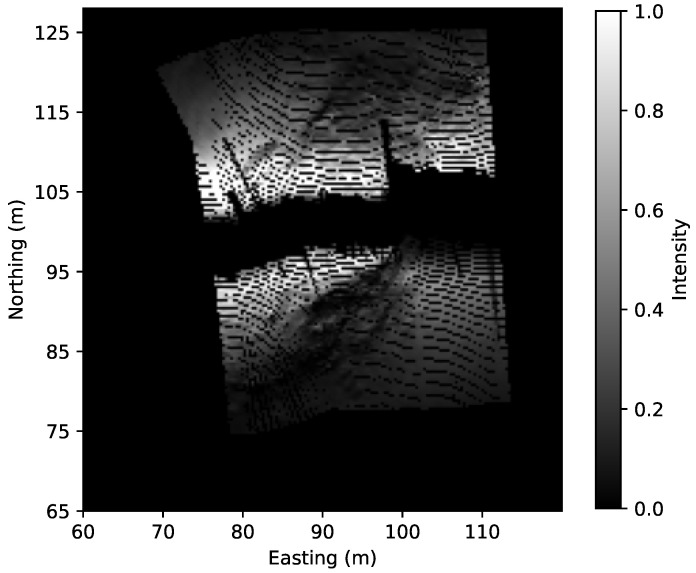
Georeferenced SSS of Figure 14b using the estimated bathymetry.

**Figure 16 sensors-22-05092-f016:**
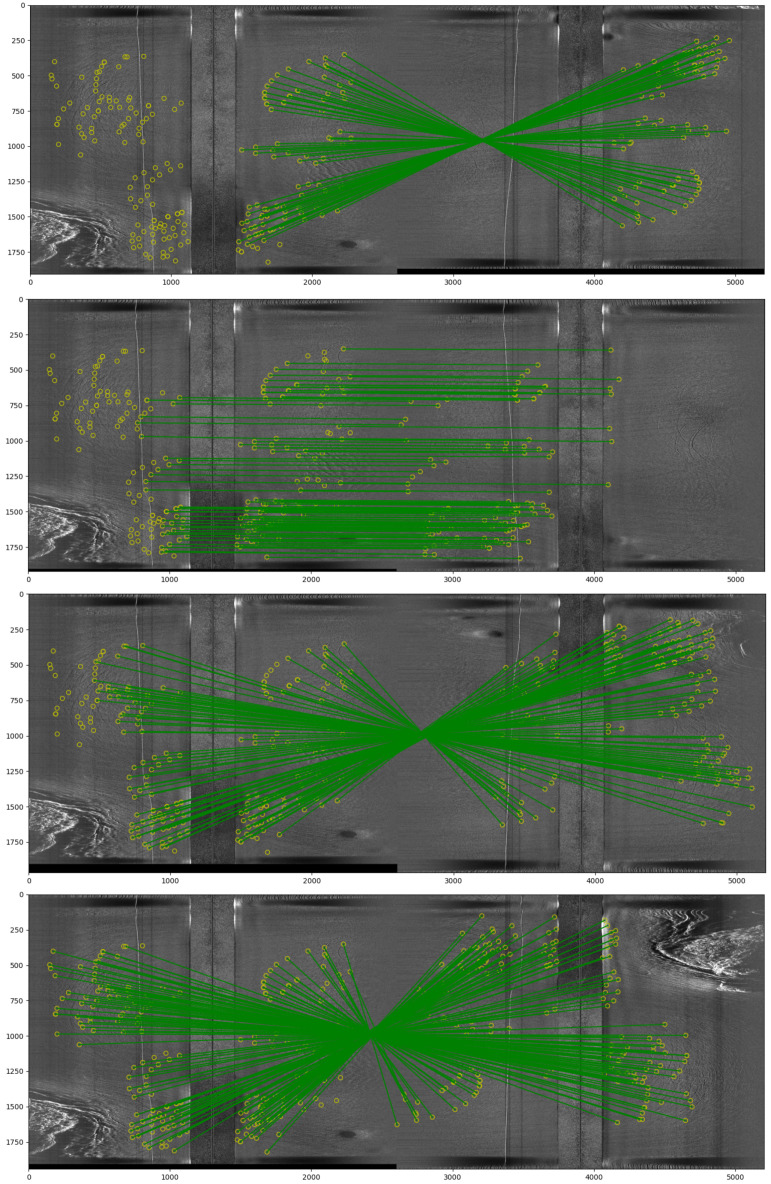
An example of how sidescan waterfall images can have features matched between them that are later used in a SLAM framework to both correct the navigation and constrain the seafloor depth estimate at these points. This is an example of where we hand annotated the matching features. (The features need a much higher resolution than here to actually see.) Each image comprises two waterfall images from different parallel lines of a survey. The same base waterfall image is shown to the left in each image. Matching waterfall images are shown to the right. The green lines connecting matching features form a crossing pattern when the AUV was moving in opposite directions and they are parallel when when moving in the same direction. As this was a lawn mower pattern survey every other line is in the opposite direction. the bottom two images are the closest two lines to the base line. The bottom line was to the starboard side of the base line (the vehicle travels downward in these images). The other three are to the port (thanks to Li Ling for these images).

**Figure 17 sensors-22-05092-f017:**
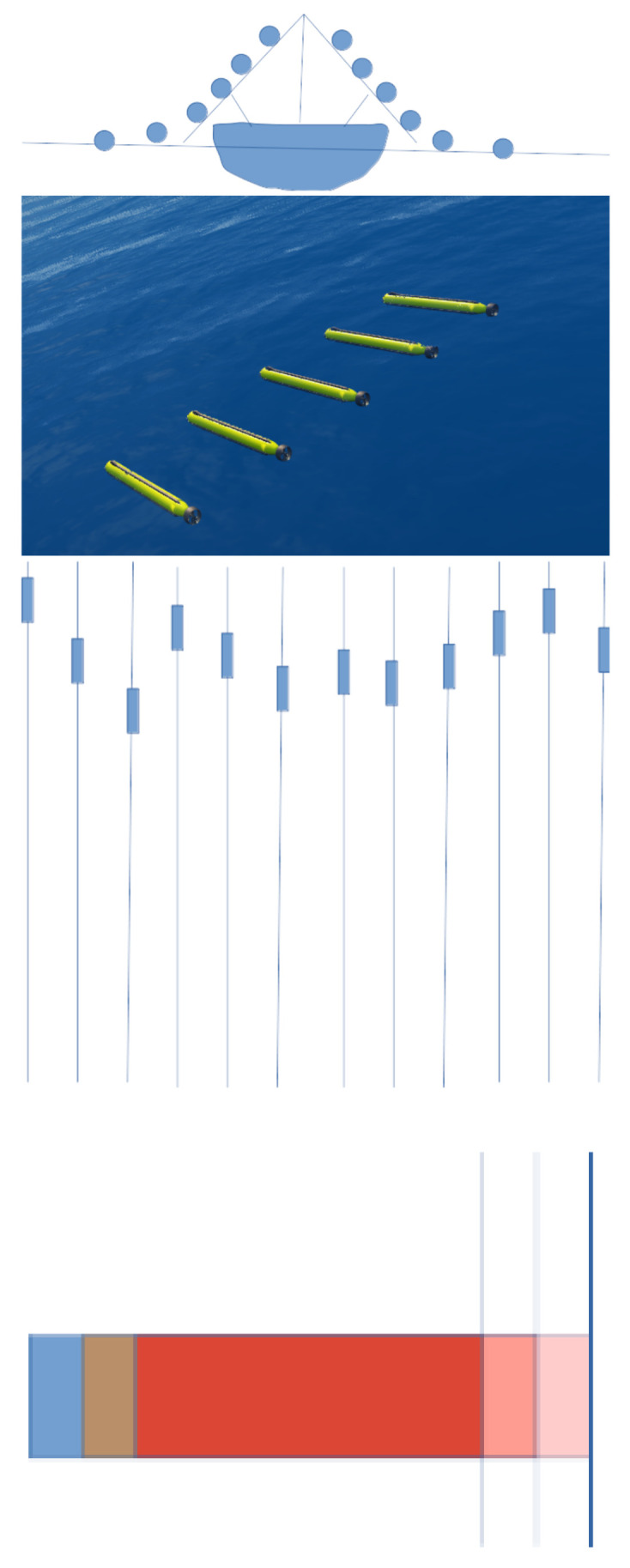
A schematic of a multi-agent small AUV fleet being launched (rather whimsically) from an automatic launcher (**top**). These would be equipped with sidescan sonar and navigation sensors. Optionally also acoustic modems that allowed them to ping one another according to some timing protocol that allowed ranges between individual AUVs to be periodically measured. The AUVs carry out the survey in parallel lines (**middle**). Finally we illustrate the resulting starboard side-scan swathes from the first three lines showing how there can be substantial overlap (**bottom**). Post mission any between-AUV range measurements can be combined with constraints formed by matching features in overlapping sonar regions to improve the dead-reckoning estimates. Once that has been performed the method of neural rendering presented here could be used to estimate the bathymetry. The rendering optimization could even be extended to refining the AUV attitude estimates. Such a system would allow rapid bathymetric surveys to be carried out from small boats with low cost equipment.

**Table 1 sensors-22-05092-t001:** A comparison between different methods of bathymetry reconstruction from sidescan.

Extra Bathymetric Data Required	Extra Bathymetric Data Not Required	Lambertian Scattering Model	Data-Driven Scattering Model
[8,9,10,11,12,13,14]	[3,4,15,16,17]	[3,4,8,9,10,12,14,15,16,17]	[11,13]

**Table 2 sensors-22-05092-t002:** The main inputs and outputs of our method.

Inputs	Description
$o_{i}\in R^{3}$	sidescan origin at ping *i*
$R_{i}\in SO\left(3\right)$	sidescan rotation matrix at ping *i*
$I_{i,n}\in R$	intensity at ping *i* and time bin *n*
**Outputs**	
$\theta :R^{2}\to R$	bathymetry height map
$R:R^{2}\to R_{+}$	albedo map
$\Phi :R\to R_{+}$	beam profile
$A_{i}$	per-survey line gain

**Table 3 sensors-22-05092-t003:** Dataset and survey area characteristics.

Property	Value
Bathymetry resolution	0.5 m
Sidescan type	Edgetech 4200 MP
Sidescan range	0.035 s ⇒~50 m
Sidescan frequency	900 kHz
Composition	~70% Sedimentary rock, 30% sand
Mean altitude	17 m
Survey area	~350 m × 300 m
Sidescan pings	~93,000

**Table 4 sensors-22-05092-t004:** Statistics on the errors of the estimated bathymetry.

	Max (m)	Min (m)	Mean (m)	Abs. Mean (m)	STD (± m)
ours	2.458	−0.955	0.097	0.195	0.228
[12]	1.166	−1.244	−0.006	0.028	0.065
[16]	0.47	−0.54	0.00	-	0.12

**Table 5 sensors-22-05092-t005:** Quantitative results.

	MAE (m)	Cosine Similarity
$C=e^{-10}$	0.195	0.817
$\;C=e^{-3}$	0.214	0.814
$C=e^{-1.5}$	0.605	0.799
$C=e^{-1}$	0.893	0.791
$C=e^{-0.5}$	0.556	0.800
C=1	0.207	0.815

## Data Availability

Data sharing is not applicable to this article.

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
