# Peer review of "Sidescan Only Neural Bathymetry from Large-Scale Survey"

_sensors, 2022, doi:10.3390/s22145092_

Round 1

Reviewer 1 Report

In this manuscript, the authors proposed a bathymetry reconstruction method of traditional side scan sonar data using the deep learning model (SIREN). The method was verified by the measured data and showed potential engineering application value. However, some issues should be taken into consideration.

Bathymetry terrain reconstruction of side scan sonar data does not seem to be a popular research topic, which can be seen from the manuscript references. Some of the references are either very old or in submitted state by the authors, and there are very few relevant studies.

During the side scan sonar measurement, the seabed scattering patterns are not only affected by the terrain but also can be affected by the seabed sediments. The seabed scattering patterns of different seabed sediments can be quite different. Because it is hard to judge whether the echo intensity change is caused by the terrain change or the sediment change, the sediment problem is an important issue in terrain reconstruction of side scan sonar data. However, the manuscript does not have any mention.

New generations side scan sonars (like EdgeTech 6205s) already have the bathymetric abilities using multi-phase technology, and can simultaneously measure side-scan images and seafloor topography. However, the manuscript also does not have any mention or make any comparison.

The manuscript cites four unpublished (submitted) works of the authors', which have not been peer-reviewed and whose methods and conclusions have not been confirmed. These unpublished (submitted) works should not be used as references.

Almost in all parts of the manuscript, the theories, parameters or models were derived from the authors’ unpublished papers. But there is no result comparison between the current work and previous works. How can the authors prove that this manuscript is innovative enough to be a new article compared to these unpublished works?

The analysis of experimental results is not very convincing without a complete results presentation. And the method in this manuscript does not make any comparison with other methods. How can the authors prove the innovation of the method? Also, the accuracy assessment method is not very reasonable.

Other comments:

Introduction:the author mainly introduces the differences between your method and relevant methods, but there are few descriptions of the relevant research, therefore references in the manuscript are quite few. The authors do not mention any researches of seabed sediment problem or new generation side scan sonars.

Figures:

In Figure2, what do the vertical and horizontal coordinates means? How do the gray scales represent the side scan echo strengths? What features were used for image point matching. It is recommended to label the different graphs and explain the differences in detail. Whether all the yellow points are match points?

In Figure 3, What do the different colors mean? Is the caption description closely related to the image? What is the overlapping rate of adjacent side scan images?

In Figure 4 and Figure 6, the font styles and sizes are not uniform.

Are Figure 7 and Figure 8 in the correct sequence? In Figure 7, what does gray scales represent?

In Figure 10, the color bars of these terrain gradients are missing.

Equation: Please check whether are all letters in the formula explained in detail.

Methods:

The manuscript introduces the deep learning representation model by citing the unpublished author’s work. Is it exactly the same as author’s previous works?

The downsampling would cause data loss. How do you solve this problem?

How do the authors deal with the gap between port and starboard side scan images?

The method section lacks of data accuracy assessment part.

Experiments & Results:

There was only one waterfall images of the side scan sonar. The side-scan image of the full survey area was not shown in the manuscript.

Line 264:“However, there are many small rocks near the ridge that are not reconstructed” There are also rocks far from the ridge which are not reconstructed.

It is not reasonable to assess the bathymetric terrain only using the average error and cosine similarity, because the maximum errors are more important for the shipping safety.

The author does not compare the proposed method with other method. The results should be compared with previous works.

Does the C in Table 4 mean the constant C? How the constant is changeable?

Discussion:

The shadows should not be neglected. The authors should explain how does the proposed method deal with the large-area shadows.

The downsampling operation will remove not only noise but also important data. How do the authors solve the potential problems?

References:

These unpublished (submitted) works should not be used as references.

Reviewer 2 Report

The authors present an interesting methodology to fit bathymetry and albedo from a set of side-scan sonar images (a.k.a. lines). The method only makes the physically reasonable assumption of a Lambertian backscattering model (and uses reasonable numerical "tricks" to fit data at nadir), and then uses survey information to retrieve all other elements in the side-scan-sonar image formation model (from beamform to albedo) by neural network optimization.

The methodology and the results are nicely presented in sections 3-5, and comparison with a multibeam ground truth of the same area is amazingly good. I consider the manuscript worth of being published in Sensors.

However, there are some important issues with the presentation of the article. That motivate my decision of having it "major revised".

In particular, the introduction is not an introduction to the problem, and section 2 seems to be just out of place. In particular, the last paragraph of section 1 reads "The main contribution here is the proposed method of using smaller AUV platforms with relatively simple equipment to carry out high quality bathymetric surveys". However, this is never addressed in this article (a different one?), although admittedly could be a potential application of the method that is worked out in sections 3 to 5.

The same happens with section 6, which is not a discussion, but a list of unsolved (or unaddressed) problems of the method. I encourage the authors to discuss their methodology (paragraphs 3-4 of current section 1 are good candidates) and their results, and perhaps propose here how their methodology could be applied to develop bathymetries using swarms of interconnectad UAVs (current section 2).

There are also some important issues with the article background, in particular with the continued references by the authors to their previous works: up to 4 of their most recent references that happen to be, as of today, still "submitted", i.e. unpublished. Hence the authors must provide more in depth information in the current article, instead of refering the reader to unavailable sources (they appear not to exist even as preprints). In particular, the authors should explain more in detail the cosine-squared approximation, and the estimation of beamform (a.k.a. bidirectional directivity function) and albedo for each side-scan line; regarding the optimization procedure, they could motivate the use of an implicit neural network, instead of other explicit or implicit surface representation functions (e.g. radial basis functions), and should expand on how global estimation of K is done to improve convergence, and how is the interplay between backpropagation and gradient descent; regarding the training of the MLP, more details are needed, such as the data partition into training and validation sets (if it is already described, e.g. those 100 side-scan pings, it is not clear enough); regarding the validation of their results with respect to MB data, the authors should explain more in detail the "cosine similarity" measure they used in unpublished reference [6].

Those are the main issues with the manuscript. However, I would like the authors to address or clarify also the following points:

L77 and ff. The authors use the term "pose" instead of the more common one "attitude" to refer to side-scan orientation (needed for image projection onto the bathymetric surface). Are they equivalent as I understand or do "pose" include other information?

L231 and ff. The authors describe their data set as having "52 surveying lines"; are these 52 images of the surveyed area? About the 900 kHz sonar frequency, I understand that is the transducer frequency (really high, unpractical for deeper areas I would be more interested in), and not the sampling frequency (bins per second), and I would like to know how relevant is that high frequency to the good performance of the algorithm (if the authors have considered that issue).

L274 and ff. The authors say that they set beamform to a constant for angles below 10º, and that improves convergence. Could they elaborate on this, because I do not see why it should; moreover, why not too fix to a constant the beamform for angles above 80º?

L317. The authors summarize their method as a "neural shape-from-shading method", but it actually is an enriched shape-from-shadow, as it includes information about first return echo depth, which helps calibrate the surface model.

L319. The authors highlight that the method can "fuse multiple lines of side-scan data", but that is taken somewhat as granted in the methodology (perhaps again explained in unpublised reference [6]?).

L329. The authors propose as an improvement the inclusion of SSS "pose" (attitude data) "into optimization jointly": what do they mean? Angular data had already been taken into account during optimization.

Reviewer 3 Report

In this paper, the authors propose a method of reconstructing bathymetry using only sidescan data from large-scale surveys by formulating the problem as a global optimization. In general, the work in this paper is well described. However, some useful discussions should be performed before acceptance, as these discussions would guide the readers’ research.

1. The Interferometric synthetic aperture sonar [C1] can also provide the Bathymetry. However, the authors do not discuss related method in section 1. The authors should comprehensively discuss this method in section 1.

[C1] R. Hansen, et al. Interferometric synthetic aperture sonar in pipeline inspection, IEEE Oceans Conference, Doi: 10.1109/OCEANSSYD.2010.5603518

2. In section 2.1, the authors discuss the noise in SSS data. Nowadays, the non-Gaussian noise like Class A[C2], alpha noise [C3] is usually discussed in shallow water. The authors should discuss this in section 2.1

[C2]Zhang,et al.Parameter Estimation for Class A Modeled Ocean Ambient Noise.Journal of Engineering and Technological Sciences,Doi: DOI: 10.5614/j.eng.technol.sci.2018.50.3.2.

[C3]Mahmood, et al, “Modeling Colored Impulsive Noise by Markov Chains and Alpha-Stable Processes,” in OCEANS 2015 MTS/IEEE, (Genoa, Italy), Doi: 10.1109/OCEANS-Genova.2015.7271550.

3. In section 5, the authors should directly compare their method to existed method in [6], as the authors’ method is just the extension of [6].

Round 2

Reviewer 1 Report

In the revise manuscript, the authors have response all of the comments. Most of the responses are acceptable. However, some issues still should be taken into consideration.

Table 4: Do your method have largest errors than other methods? How could this result prove the advantage of your method?

Appendix B: The original suggestion means that author should show the geocoding side scan image of the whole survey area if possible. Added side scan images were quite similar.  

I suggest that authors could add your responses into the discussion section about problems caused by shadows and down-sampling operations.

Reviewer 2 Report

The authors have done a great job, both rearranging their manuscript, and answering all of my concerns. I recommend the article for its publication with just some very minor suggested changes:

L33: Please include more information about Edgetech 6205 and Klein HydroChart 3500, e.g. manufacturer and country.

L34: Delete the "but" at the end of the line.

L36: Change "depth less than" to "less deep than".

L37: Change "more noisy than" to "noisier than".

L37: By "many outliers needed to be further addressed", do you mean "many outliers that must be filtered out"?

L38: Perhaps start with "However, in this work we focus on using non-interferometric SSS because they are ubiquitous..."

L132: Please introduce the acronym INR (used below) here, after "implicit neural representation". [Despite the abbreviation table at the end, I find it easier to find them also defined in the text, close to their first use.]

L189: "can help/improve/accelerate convergence".

L191: Eq. (1): Please, clarify the meaning of the single and double vertical bars in the expression.

L211, L283, L372, L402: Please replace sonar "poses" with "position and orientation" or with "attitude", as explained in your previous response.

L217: "open-source software" is more common.

L316: "due to there being few shadows in our dataset specifically", or simpler "due to the few shadows specifically contained in our dataset".

L367-368: "The neural rendering based methods ... have the promise ..."

L403: "instead of treating them as a given" (instead of "as true")

Reviewer 3 Report

The authors well addressed the review's comments.

Author Response

Thank you!